# Unsupervised detection and fitness estimation of emerging SARS-CoV-2 variants: Application to wastewater samples (ANRS0160)

Alexandra Lefebvre[1,2]*, Vincent Maréchal[3], Arnaud Gloaguen[4], The Obépine Consortium[5‡], Amaury Lambert[2,6◉], Yvon Maday[1◉]

**1** Sorbonne Université, Université Paris Cité, CNRS, Laboratoire Jacques-Louis Lions, LJLL, Paris, France, **2** Stochastic Models for the Inference of Life Evolution (SMILE), Center for Interdisciplinary Research in Biology (CIRB), Collège de France, CNRS UMR, INSERM U1050, PSL Université, Paris, France, **3** Sorbonne Université, INSERM, Centre de Recherche Saint-Antoine (CRSA), UMR_S, Paris, France, **4** Centre National de Recherche en Génomique Humaine, Institut de Biologie François Jacob, CEA, Université Paris-Saclay, Évry, France, **5** Faculté des sciences, Sorbonne Université, Paris, France, **6** Institut de Biologie de l'ENS (IBENS), École Normale Supérieure, CNRS UMR, INSERM U1024, PSL Université, Paris, France

◉ These authors contributed equally to this work.
‡ Membership list can be found in the Acknowledgments section.
* alexandra.lefebvre@math.cnrs.fr

## Abstract

Repeated waves of emerging variants during the SARS-CoV-2 pandemics have highlighted the urge of collecting longitudinal genomic data and developing statistical methods based on time series analyses for detecting new threatening lineages and estimating their fitness early in time. Most models study the evolution of the prevalence of particular lineages over time and require a prior classification of sequences into lineages which is prone to induce delays and biases. More recently, several authors studied the evolution of the prevalence of mutations over time with alternative clustering approaches, avoiding specific lineage classification. Most existing methods are either non parametric or unsuited to pooled data characterizing, for instance, wastewater samples. The analysis of wastewater samples has recently been pointed out as a valuable complementary approach to clinical sample analysis, however the pooled nature of the data involves specific statistical challenges. In this context, we propose an alternative unsupervised method for clustering mutations according to their frequency trajectory over time and estimating group fitness from time series of pooled mutation prevalence data. Our model is a mixture of observed count data and latent group assignment and we use the expectation-maximization algorithm for model selection and parameter estimation. The application of our method to time series of SARS-CoV-2 sequencing data collected from wastewater treatment plants in France from October 2020 to April 2021 shows its ability to agnostically group mutations in a consistent way with lineages B.1.160, Alpha, B.1.177, Beta, and with selection coefficient estimates per group in coherence with the viral dynamics in France reported by Nextstrain. Moreover, our method detected the Alpha variant

**Data availability statement:** The data analyzed in this article are available at the link: https://data-dataref.ifremer.fr/bioinfo/ifremer/obepine/lsem/data/dna-sequence-raw/. The code related to this article is provided at the link: https://github.com/AlexandraLefe/FT-mixture.

**Funding:** This work was supported by ANRS-MIE (EmerEaUde project ANRS0160 received by VM and Jean-François Deleuze (Centre National de Recherche en Génomique Humaine (CNRGH), Institut de Biologie François Jacob, CEA, Université Paris-Saclay). This work was also funded by the French National Research Agency (ANR) under the France 2030 programme (reference ANR-24-MIEM-0004 received by VM and YM as part of the OBEPINE+ platform). ALe received salary from both funders. The funders had no role in study design, data collection and analysis, decision to publish, or preparation of the manuscript.

**Competing interests:** The authors have declared that no competing interests exist.

as threatening as early as supervised methods (which track specific mutations over time) with the noticeable difference that, since unsupervised, it does not require any prior information on the set of mutations.

## Author summary

The SARS-CoV-2 pandemics has been characterized by successive waves of emerging variants replacing previously dominant ones. A variant is characterized by a combination of mutations, with some mutations possibly shared among variant relatives. The early detection of emerging variants is of great importance in order to adapt public health responses to viral evolution. Wastewater surveillance has been highlighted as a valuable complementary approach to clinical sample analysis mostly because it is representative of the viral circulation at a population level. Indeed, all infected individuals, wether symptomatic or not, contribute to wastewater samples. Wastewater surveillance however is subject to some statistical challenges as the viral genetic material is highly fragmented, incomplete and comes from multiple individuals. In this work we propose a method, suited for wastewater samples, grouping viral mutations according to their frequency trajectory through time in an agnostic manner and we detect threatening variants without prior knowledge on their characteristic mutations as early as methods targeting known specific mutations.

## Introduction

The Covid-19 pandemic has been characterized by the successive emergence of new SARS-CoV-2 variants leading to several temporal waves and prompting the World Health Organization to classify certain variants as Variants Of Concern (VOC), variants of interest or variants under monitoring, according to their biological characteristics in terms of transmissibility, virulence, immune evasion, etc. Detecting variants of potential threat early in time is of great importance for an appropriate and rapid adaptation of public health responses to viral evolution.

Since SARS-CoV-2 genome can be found in feces [1] and other biological fluids of infected individuals, symptomatic or not, WasteWater (WW) samples give a view of SARS-CoV-2 circulation at a population level, as all infected individuals contribute to the sampling. The interest of WW surveillance has been highlighted in numerous previous studies [2–6] as a complementary approach to clinical sample analysis, as it is cost effective and it can provide knowledge about the current trend of the epidemic in the global population. The statistical analysis of WW samples is challenged by their pooled nature as they contain a mixture of fragmented sequences each associated with potentially several lineages and secreted by multiple infected individuals. Moreover, most WW sequences are incomplete. On the opposite, most clinical samples are individual-specific and contain a dominant lineage represented by almost complete sequences with limited variability, except during persistent infections [7]. The first statistical methods applied to WW samples were supervised in the sense that

they track known variants using either digital or RT-qPCR [8–13] or high-throughput sequencing data [13–17]. They are therefore not suited for detecting newly emerging (also called cryptic) variants. Some alternative approaches based on the amplification of small and specific regions of SARS-CoV-2 genomes extracted from WW samples revealed linked polymorphisms [18,19]. These methods however are limited to targeted genomic segments and seem to provide only polymorphism counting and frequency outputs with no fitness estimates.

The genomic surveillance of COVID pandemic trough time provided a considerable amount of longitudinal genomic data and favored the development of statistical methods for time series analysis applied to the detection of emerging variants and the estimation of their selective advantage. There currently exists two main types of approaches. The first type relies on analyzing the prevalence through time of particular lineages [20–27] and offer statistical methods to estimate the relative fitness of lineages. Such methods require a prior clustering of sequences into lineages using, for most of them, Pango lineages [28] or alternative phylogenetic methods which are prone to induce delays and misinterpretation in particular for newly emerging variants. Moreover such methods are unsuited to pooled samples as they require one sample to be associated with one viral sequence. The second type of methods relies on analyzing the prevalence through time of mutations and includes some methods both designed for clinical and pooled samples. These methods display a variety of clustering strategies including the k-medoids partitioning [29], a weighted mutation network [30], the Levenshtein distance between sequences [31], latent epidemiological variables [32] or latent population genetic structure [33].

In this work we propose an alternative unsupervised method that falls into that second category for clustering mutations according to their frequency trajectories over time and estimating cluster fitness from time series of pooled mutation count data. As our parameter to estimate is our clustering criteria itself, we take advantage of the statistical power gain of both clustering and estimating fitness at once. Our model is suited for pooled data and therefore it is particularly useful for analyzing WW samples although it can be applied to an aggregation of clinical samples. In this paper, we start with a presentation of our method with its mathematical design, the statistical methods we used before applying it to a variety of simulated datasets to present its strengths and weaknesses. We assess our model over two WasteWater Treatment Plants (WWTP) datasets collected in Nantes, France, during the emergence of B.1.1.7 (Alpha) and B.1.351 (Beta), the decline of B.1 and the transition of B.1.177 VOC. We demonstrate its ability to group mutations in accordance with the retrospective viral dynamics in France at the time of data collection reported by Nextstrain project (https://nextstrain.org), notably for lineages B.1.160, Alpha, B.1.177 and Beta. We also show its capacity to detect the Alpha variant as threatening as early as supervised methods with the noticeable difference that, since unsupervised, it does not require any prior knowledge on mutations. We finally discuss the limits of our model and propose perspectives.

In order to avoid any confusion between the term *cluster* in statistical clustering and *cluster* in epidemiology, we will use the term *group* for the former in the following.

## Method

### Mathematical modeling

Our model is a mixture of observed count data and latent group assignment variables. We consider one group not under selection named the neutral group and $K \geq 0$ groups under strictly positive or negative selection, named non-neutral groups. Group number $k \in \{0, \dots, K\}$ will be denoted $G_k$, where $G_0$ stands for the neutral group. Let $n$ be the number of mutations in the dataset, compared to a reference sequence, we introduce $Z = \{Z_1, \dots, Z_n\} \in \{0, \dots, K\}^n$ where $Z_i$ denotes the group assignment for mutation $i \in \{1, \dots, n\}$. The neutral group is associated to value 0 such that $\{Z_i = 0\}$ means mutation $i$ belongs to the neutral group.

Sequencing data are collected at $m + 1$ increasing time points $t_0, \dots, t_m$ leading to time series of mutation count data and read depths. We denote by $\mathcal{T} = (t_0 - t_0 = 0, t_1 - t_0, \dots, t_m - t_0 = T)$ the vector of differences, usually in days, between sampling date and date of origin $t_0$. Let $X_{i,t}$ be the mutation number $i \in \{1, \dots, n\}$ count at time $t \in \mathcal{T}$ and $d_{i,t}$ be the read depth at related genome position at time $t \in \mathcal{T}$. We assume a multinomial distribution for the latent variables with parameter

$\pi = \{\pi_0, \ldots, \pi_K\}$ such that, for $i \in \{1, \ldots, n\}$, for $k \in \{0, \ldots, K\}$,

$$\mathbb{P}(Z_i = k) = \pi_k \qquad \text{where} \qquad \sum_{k=0}^{K} \pi_k = 1.$$

We assume a generalized linear model with a binomial family for modeling the distribution of mutation counts conditional on group assignment as follows. For all $k \neq 0$, mutation $i$ count at time $t \in \mathcal{T}$ conditional on $\{Z_i = k\}$ follows a binomial distribution with a logit link function such that

$$\{X_{i,t} \mid Z_i = k, k \neq 0\} \sim \text{Binomial}\left(d_{i,t}, \text{Logistic}(\mu_k + s_k t)\right)$$

where Logistic is the standard logistic function, that is, for $v \in \mathbb{R}$, $\text{Logistic}(v) = e^v/(1 + e^v)$, $\mu_k \in \mathbb{R}$ and $s_k \in \mathbb{R}^\star$ are respectively the intercept (the logit of the frequency at time origin $t_0$) and the selection coefficient associated with group $G_k$, $k \neq 0$. We assume no recombination event and a constant selection coefficient over the time period $[t_0, t_m]$. Note also that a selection coefficient is a direct estimate of the slope of the trajectory conditional on group assignment with no distinction between evolutionary or epidemiological parameters. As most mutations are neutral (not under selection, constant frequency trajectory) and because we have a particular interest in groups under positive (or negative) selection, mutation $i$ count conditional on $\{Z_i = 0\}$ follows a beta-binomial distribution in order to absorb the variability of mutation frequencies at time origin for the neutral group. Therefore for all $i \in \{1, \ldots, n\}$ and $t \in \mathcal{T}$, we assume that

$$\{X_{i,t} \mid Z_i = 0\} \sim \text{Binomial}\left(d_{i,t}, u\right) \qquad \text{with} \qquad u \sim \text{Beta}(\alpha, \beta)$$

where $\text{Beta}(\alpha, \beta)$ is the beta distribution of parameters $\alpha$ and $\beta$.

Let $\theta = (\pi, \mu, s, \alpha, \beta)$ be the set of parameters where $\pi = (\pi_0, \ldots, \pi_K) \in [0, 1]^{K+1}$ such that $\sum_{k=0}^{K} \pi_k = 1$, $\mu = (\mu_1, \ldots, \mu_K) \in \mathbb{R}^K$, $s = (s_1, \ldots, s_K) \in (\mathbb{R}^*)^K$ and $\alpha, \beta \in (0, \infty)$, let $Z = \{Z_1, \ldots, Z_n\}$ and $X = \{X_1, \ldots, X_n\}$ where, for $i \in \{1, \ldots, n\}$, $X_i = (X_{i,t})_{t=(0,\ldots,T)}$, the joint probability of $X$ and $Z$ writes

$$\mathbb{P}(X, Z \mid \theta) = \prod_{i=1}^{n} \mathbb{P}(Z_i \mid \pi) \prod_{t \in \mathcal{T}} \mathbb{P}(X_{i,t} \mid Z_i; \mu, s, \alpha, \beta). \tag{1}$$

## Statistical tools

**Clustering and parameter estimation.** We use the expectation-maximization (EM) algorithm [34] for clustering and parameter estimation with details of the process provided in Supporting information S1 Text.

**Selection of the number of groups.** The selection of the number of groups is a delicate task in unsupervised clustering. The number of groups is usually chosen as the one that minimizes an Information criteria such as the Bayesian Information Criterion (BIC) or Integrated Complete-data Likelihood (ICL) [35]. However, when applied to real data, it often happens that neither the BIC nor the ICL reaches a minimal value for an increasing number of groups. The elbow method applied to the log-likelihood is a common alternative option when a clear elbow is displayed. If none of these methods are applicable, alternative strategies are empirically tested and motivated. In our context, we propose the following alternative option based on a lower limit of group sizes, in terms of number of mutations. We assign a group to a mutation using the maximum a posteriori (MAP) of group assignment, that is $\arg\max_{k \in \{0,\ldots,K\}} \mathbb{P}(Z_i = k \mid X_i = x_i)$ where $(x_i)$ denotes the observed vector of mutation counts. Denoting $N_k$, the number of mutations composing the smallest group conditional on $K = k$ non-neutral groups, we propose, through our empirical tests over diverse WWTP datasets, to select the number of non-neutral groups $K$ such that $K = \arg\max_k \{N_k \geq 5\}$ (respectively $K = \arg\max_k \{N_k \geq 3\}$ and $K = \arg\max_k \{N_k \geq 2\}$) for

analyses covering time periods in the order of months (respectively weeks and days). In other words, we select the highest value for $K$ such that all groups contain at least five, three or two mutations according to the length of the studied time period. The upper limit $K_{max}$ for tested $K$ is such that $N_{K_{max}-2} = N_{K_{max}-1} = N_{K_{max}} = 1$. In other words, we pursue the analysis conditional on an increasing number of non-neutral groups $K$ until one group, at least, contains only one mutation for three consecutive tested values for $K$. Let us stress the fact that the aforementioned values as lower group size limit for selecting $K$ and for choosing $K_{max}$ were empirically determined over four WWTP SARS-CoV-2 datasets presented in [6] and [36] and should be adapted to the context of other viruses, clinical versus WW samples, different time periods, various sampling effort and dataset sizes, etc. Future work and investigations are in progress in order to empirically lift general conditions of application.

## Model assessment over simulated datasets

In order to evaluate to which extent the number of observations and/or the strength of the signal, in terms of the range of parameter values, influences the performances of our model, we began its assessment over datasets simulated with the same model. We provide, in this paragraph, a summary of the main results and we refer the reader to Supporting information S2 Text for more details, illustrations and comments on a selection of simulated studies. Let us start by reminding that the parameter estimator being the maximum likelihood estimator, it is biased with a limited number of observations. We noticed through various simulated studies, that the bias of the estimator induced by a limited number of time points (2 time points) and/or read depths (mean below 10) is negligible, in contrast to the bias induced by a limited number of mutations ($n = 25$). However, in the framework of our simulation schemes, the bias becomes negligible from $n = 100$ for all parameter estimates.

We also note the very accurate posterior group assignment with AUCs first quartiles above 0.95 in each simulation scheme except for limited read depths (mean below 10). This result comforted our choice of using the MAP of group assignment for assigning a group to a mutation as well as applying a threshold for read depths during the preparation of WWTP datasets.

Finally, we empirically tested the impact on our results of the assumption of conditional independence of mutation counts through time. Returning to the main equation of our model, Eq (1), we see that mutation counts at different time points are assumed to be independent conditional on group assignment (but not independent without that conditioning). This assumption implies that we ignore the temporal dependency structure of time series. As previously noted by several authors [37–39], a common way for modeling an evolutionary process is a hidden Markov model with an underlying Wright-Fisher diffusion process. In order to better capture the structure dependency of time series, a natural extension of our model could consist of a hidden random walk conditional on group assignment, that is, a random walk with Gaussian transition probabilities for modeling the latent parameters of our binomial emission probabilities followed by observed mutation counts. That model will be called *the hidden random walk model* in the following.

Such a model better captures the dependency structure of our data, however it is computationally intensive to fit and its in-depth study is in progress for a future work. Nevertheless, we can use it for simulating datasets to analyze with our model. We refer the reader to Supporting information S3 Text for a description of the mathematical modeling of the *hidden random walk model* and a presentation and interpretation of our results. As a brief summary, we noticed that the assumption of independence of mutation counts through time conditional on group assignment, induces a bias when estimating most parameters with our model. However, the parameter of highest interest, that is the vector of selection coefficients $s$, seems less impacted by such assumption.

## Mutation profile matrix

In order to a posteriori verify our outputs over real datasets, we will use a mutation profile matrix defined as a mutation $\times$ lineage matrix filled with the probability for each mutation to belong to a known lineage. We followed the procedure

provided by Virpool [40] to compute this matrix using script `src/gisaid/process_gisaid.pl` with GISAID sequences collected between 2020-01-01 and 2021-05-15 and default parameter 'MAX PER MONTH = 50000'; 'MIN LENGTH = 25000' for a random sampling of sequences and script `src/profile_estimation.py` for the inference. Both scripts are available at https://github.com/fmfi-compbio/virpool. We extracted lineages B.1, B.1.1, B.1.160, B.1.1.7 (Alpha), B.1.177 and B.1.351 (Beta) from VirPool's outputs, for them to be the main circulating lineages at the time of the analysis, and we computed the frequency of each mutation observed at least once for at least one of these lineages.

## Results

We present in this section a collection of results obtained from two WWTP datasets collected in Nantes (France) and presented in [36]. After a presentation of the datasets, we start by considering the whole set of time points, from October 2020 to April 2021, for a general evaluation of the method, before restricting datasets to a selection of time points between October and December 2020, during the emergence of Alpha VOC, for an assessment of its performance in detecting (unknown) emerging variants early in time. We compare our results with the retrospective viral dynamics reported by Nextstrain and with the results of [36] who targeted specific mutations in order to verify that our outputs are consistent with those of supervised and/or retrospective approaches.

### Datasets

Time series of SARS-CoV-2 sequences collected daily or weekly from October 2020 until May 2021 from two WWTP in Nantes were downloaded in fastq format at https://data-dataref.ifremer.fr/bioinfo/ifremer/obepine/lsem/data/dna-sequence-raw/. Sample collection, data preparation and parameters set for basecalling, de-multiplexing and mapping using the nCoV-2019 sequencing protocol v3 of the ARTIC network https://artic.network with protocol https://community.artic.network/t/ncov-2019-version-3-amplicon-release/19 are detailed in [36]. The authors kindly provided us with files in .bam format. We obtained 12 samples spaced 5 to 25 days apart collected from 2020-10-20 until 2021-04-06 for the first WWTP (WWTP1) and 17 samples spaced 1 to 15 days apart collected from 2020-10-06 until 2021-04-20 for the second WWTP (WWTP2). This time period is characterized by the emergence of Alpha and Beta (mid-November 2020 and beginning of January 2021 respectively) and the decline of B.1.160 (starting around mid-October 2020) in France. Estimated frequencies of main circulating variants in France are reported by the Nextstrain project at the link https://nextstrain.org/groups/neherlab/ncov/france?d=frequencies&dhttps://nextstrain.org/groups/neherlab/ncov/france%3Fd=frequencies&f_country=France&m=div&p=full&r=division We used iVar [41] with default minimum quality threshold for sliding window to pass (default value in iVar: 20) and a zero frequency threshold for single nucleotide variants calling using the Wuhan-Hu-1/2019 as reference genome (GenBank:MN908947.3) as well as samtools depth with same quality threshold for read depth per position and we excluded indels.

### Analyses from October 2020 to April 2021

WWTP1 (respectively WWTP2) dataset contains 30,043 (respectively 36,595) mutations, most of which being of near zero frequency in all samples (i.e. at all time points). In order to avoid an accumulation of near zero frequency mutations, mostly related to sequencing errors or transient variants (recent mutations under purifying selection or in the process of stochastic extinction), mutations of frequency below 0.05 in a quarter or more of the samples were removed. Moreover, we assume that no data is available under a minimal read depth and therefore, a read depth below 10 is set to zero (along with related mutation counts). We finally obtained a dataset composed of 155 (respectively 154) mutations for WWTP1 (respectively WWTP2). Read depth quantiles at 0.25, 0.5 and 0.75 are 37, 98 and 176 (respectively 38, 93 and 161) and range from 0 to 507 (respectively from 0 to 551) for WWTP1 (respectively WWTP2). The analysis of WWTP1 (respectively WWTP2) over its whole time period will be denoted `WWTP1-2020-Oct-2021-April` (respectively `WWTP2-2020-Oct-2021-April`).

**Selection of the number of groups.** The BIC and ICL criteria of models composed of $K = 1$ to $K = 12$ non-neutral groups are represented, along with minus two times the log-likelihood ($-2 \log L$) in Fig 1. For a better visualization, we omitted values associated to $K = 0$ in the graphical representation for them to be very much higher and to offer limited information. Additionally, for each tested $K$, we reported the size of the smallest group $N_K$, in terms of number of mutations, on the top axis of each graph.

Let us remind that the BIC adds to $-2 \log L$ a penalty function of the number of parameters in the model and that the ICL adds a penalty to the BIC according to model entropy (an additional note regarding model entropy is available in Supporting information S4 Text). We notice that the BIC and the ICL decrease with an increasing number of groups up to $K = 12$ non-neutral groups and their values are almost equal to $-2 \log L$ which means that each additional group leads to such a gain in log-likelihood that penalizing the model with the number of parameters and model entropy is not sufficient to draw conclusions from Bayesian criteria. This result reflects a limit of our model in capturing sufficient variability in the datasets. We assume indeed a constant selection coefficient over the studied time period and the variability lying in binomial distributions is increased for the neutral group (beta-binomial distribution) but not for non-neutral groups (logit transformation with constant parameters). Moreover, one group is not associated to one variant stricto sensu but it is a set of mutations with similar frequency trajectory. Shared mutations among several lineages may form subgroups with their own parameter estimates.

Furthermore, it is not possible to clearly distinguish any elbow in the log-likelihood. In such context we propose the alternative option detailed in Section *Method/Statistical tools* and based on the size of the smallest group $N_K$ for selecting $K$, such that $K = \arg \max_k \{N_k \geq 5\}$ for a dataset covering several months. These choices led to analyzing WWTP1 (respectively WWTP2) dataset conditional on $K = 6$ (respectively $K = 5$) non-neutral groups.

**Clustering and parameter estimation with comparison to supervised and retrospective approaches.** In this paragraph we assume that the number of groups is selected as described in the previous section and we report and analyze quantities computed conditional on a (a priori) fixed number of groups. Estimated quantities will be denoted with a hat symbol and non-neutral groups are ordered in decreasing order of their selection coefficient. In order to lighten notation, there will be no distinct notation between groups or parameter estimates in different analyses. Consequently,

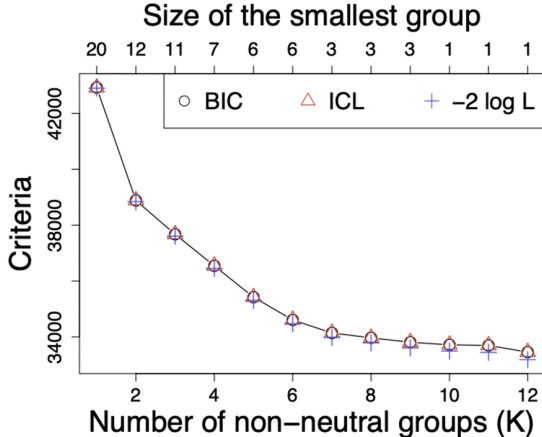
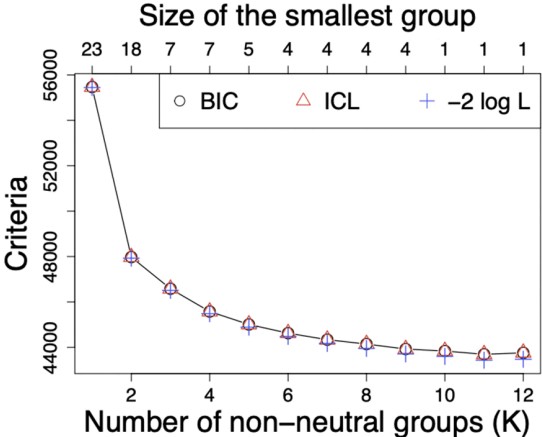

WWTP1 dataset                     WWTP2 dataset

**Fig 1**. **Selection of the number of groups in WWTP1 and WWTP2 datasets over their entire time period.** BIC, ICL along with minus two times the log-likelihood ($-2 \log L$) of models composed of $K = 1$ to $K = 12$ non-neutral groups with WWTP1 (respectively WWTP2) dataset over its whole time period, from 2020-10-20 to 2021-04-06 (respectively from 2020-10-06 to 2021-04-20) on the left (respectively right) panel. For each dataset and $K$, the size of the smallest group (in terms of number of mutations) is reported on the top axis.

different quantities may have same notation. For instance $G_1$ may refer to Group $G_1$ in any analysis, $\hat{s}_2$ may refer to selection coefficient estimates of Group $G_2$ in any analysis, etc. We computed Confidence Intervals (CI) for intercepts and selection coefficients of non-neutral groups as they are the parameters of interest. All CIs in the following are given at 95%. We therefore can determine in particular if a selection coefficient is significantly different from zero with a threshold at 2.5% with its associated CI. Note finally that time origins may vary from one analysis to the other. Times are expressed in days from the time origin of the related analysis. In the following, the estimated frequency at time origin of the neutral group is computed from estimates of $\alpha$ and $\beta$ such that $\hat{f}_0(0) = \hat{\alpha}/(\hat{\alpha} + \hat{\beta})$ and the associated intercept $\hat{\mu}_0$ is given by the logit transformation of $\hat{f}_0(0)$. Estimated frequencies at time origin for non-neutral groups are computed from estimated intercepts such that, for $k \in \{1, \ldots, K\}$, $\hat{f}_k(0) = \text{Logistic}(\hat{\mu}_k)$.

In the following, mutation profiles are those computed with Virpool package [40] as described in Section *Method/Mutation profile matrix*.

Parameter estimates, their graphical representation with resulting group frequency trajectories as well as mutation profiles stratified on MAP of group assignment are reported in Figs 2 and 3 for analyses `WWTP1-2020-Oct-2021-April` and `WWTP2-2020-Oct-2021-April` respectively. Moreover, the set of mutations composing groups of rising trajectories are listed, for both analyses, in Table 1, along with their profile for some main circulating lineages at the time of the analyses. Additionally, results of analyses `WWTP1-2020-Oct-2021-April` and `WWTP2-2020-Oct-2021-April` conditional on a lower group size limit at 3 mutations, that is conditional on $K = \arg\max_k\{N_k \geq 3\}$, are provided in Supporting information S1 Fig and S2 Fig in order to verify that that later threshold is flexible.

Similar results reported in Figs 2 and 3 are consistent with the fact that both sites, WWTP1 and WWTP2, are closely located in Nantes, France. The proportion of the neutral group $\hat{\pi}_0$ estimated at 0.67 (respectively 0.69) for WWTP1 (respectively WWTP2) is the highest among all group proportions which is consistent with the fact that most mutations are not under selection. All selection coefficients are significantly different from 0 with a threshold below 2.5% as no 95%CI contains value 0.

**Comparison with global a posteriori viral dynamics in France.** We distinguish, for both sites, two groups of high frequency at time origin, ranging from 0.71 to 0.86, and negative selection coefficients ranging from $-2.06 \times 10^{-2}$ and $-1.73 \times 10^{-2}$. These groups are exclusively composed of B.1.160 mutations except Group $G_4$ for WWTP2 composed of 11/13 B.1.160 and 2/13 B.1.177 mutations. We also distinguish, for both sites, an additional group of negative selection coefficient, $G_5$ for WWTP1 and $G_3$ for WWTP2, of lowest intercept among declining groups, containing 3/11 (respectively 4/7) B.1.177 mutations for WWTP1 (respectively WWTP2). We finally have 3 groups for WWTP1 and 2 groups for WWTP2 of low frequency at time origin ranging from $0.3 \times 10^{-2}$ to $5.8 \times 10^{-2}$ and positive selection coefficient ranging from $2.77 \times 10^{-2}$ to $4.46 \times 10^{-2}$. These groups are exclusively composed of Alpha mutations (three of them being shared either with B.1.1 or Beta) except group $G_2$ in WWTP1 which contains 4/6 mutations non associated to any VOC among B.1.1.7, B.1.1, B.1.160, B.1.177 and B.1.351 (see Table 1).

These results are consistent with the viral dynamics in France during the time period considered reported by Nextstrain with B.1.160 VOC dominating at the beginning of the time period and starting to decline while Alpha was emerging and replacing it. Moreover, the frequency trajectory of $G_5$ (respectively $G_3$) in Analysis `WWTP1-2020-Oct-2021-April-K6` (respectively `WWTP2-2020-Oct-2021-April-K5`) partly reflects the overall B.1.177 frequency evolution reported by Nextstrain around 7% at the end of October 2020, 1% by mid-March 2021 and nearly 0% by the beginning of April 2021. Our model however misses the full trajectory of B.1.177 VOC with its increased frequency between November 2020 and February 2021 but it captures its overall tendency throughout the entire time period studied. This result reflects one of the weaknesses of our method induced by the assumption of constant selection coefficients over the time period studied.

The set of Alpha mutations assigned to the neutral group is C241T, C3037T, C14408T, A23403G for both datasets as well as G28881A for WWTP1. As expected, as one emerging VOC replaces a dominant one leading to a near constant frequency trajectory of shared mutations, all these mutations are shared by Alpha, B.1.1, B.1.160, B.1.177 and B.1.351 except G28881A that is shared by Alpha and B.1.1 only.

PLOS Computational Biology

|  | Group | | | | | | |
|---|---|---|---|---|---|---|---|
|  | 0 | 1 | 2 | 3 | 4 | 5 | 6 |
| $\pi$ | 0.67 | 0.08 | 0.04 | 0.04 | 0.05 | 0.07 | 0.05 |
| $\mu$ | -1.52 | -4.28 | -5.97 | -2.79 | 1.72 | -0.86 | 0.93 |
| [95%CI] | - | [-4.33; -4.24] | [-6.04; -5.89] | [-2.82; -2.75] | [1.68; 1.76] | [-0.91; -0.81] | [0.89; 0.97] |
| $s \times 100$ | 0.00 | 3.46 | 3.04 | 2.77 | -1.73 | -1.94 | -2.06 |
| [95%CI] | - | [3.42; 3.49] | [2.99; 3.10] | [2.74; 2.81] | [-1.78; -1.69] | [-2.02; -1.86] | [-2.11; -2.00] |

(a)

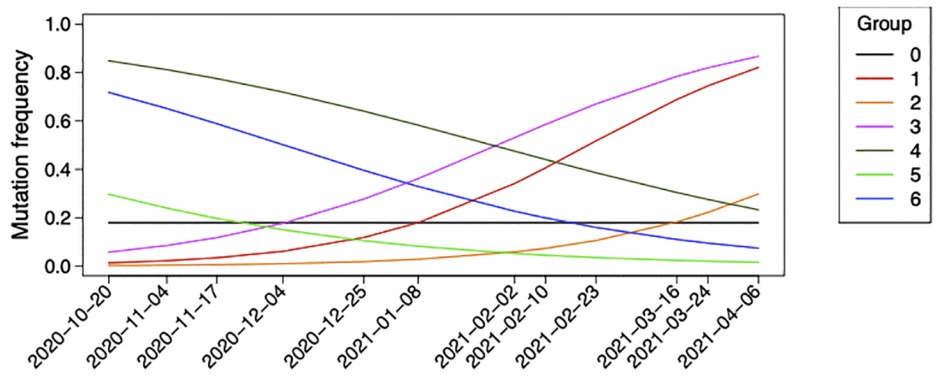

(b)

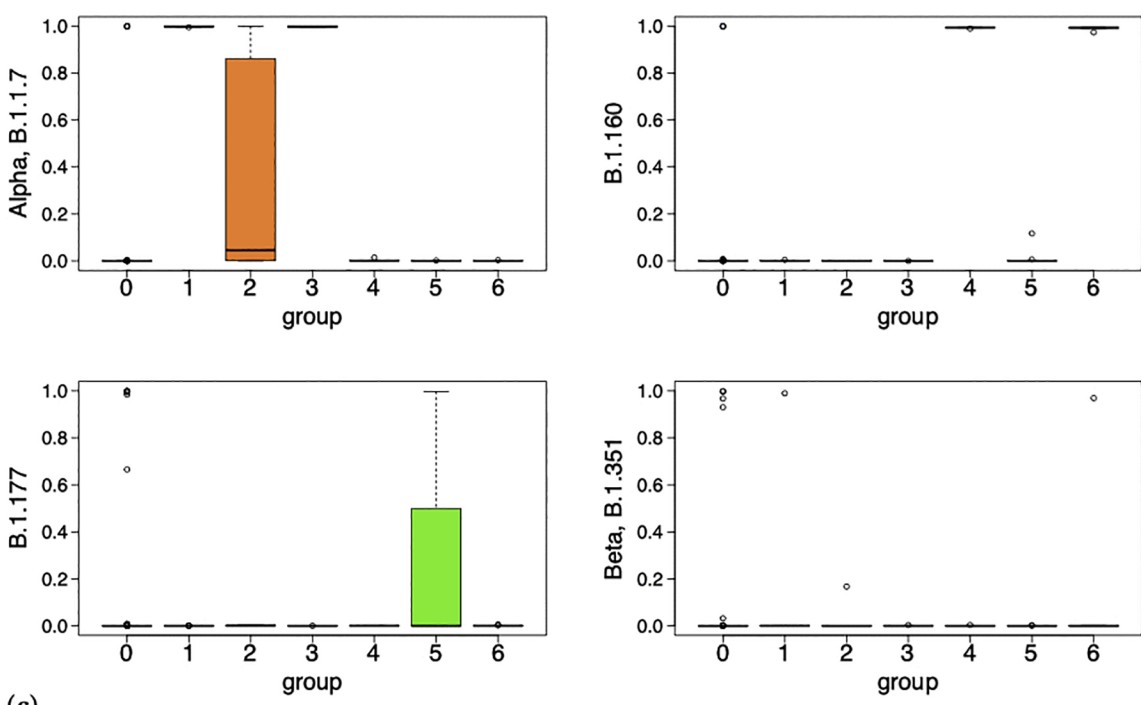

(c)

**Fig 2. Analysis of WWTP1 dataset over its entire time period** `(WWTP1-2020-Oct-2021-April)`. Parameter estimates (a), associated group frequency trajectories (b) and boxplots of mutation profile, produced with Virpool package [40], stratified on maximum a posteriori (MAP) of group assignment (c) computed conditional on WWTP1 dataset over all time points (from 2020-10-20 to 2021-04-06) and $K = 6$ non-neutral groups. Selection coefficients and their 95%CI boundaries are multiplied by 100 in (a). The estimate $\hat{\mu}_0$ in (a) is computed from estimates $\hat{\alpha} = 0.53$ and $\hat{\beta} = 2.44$.

| | Group | | | | | |
| --- | --- | --- | --- | --- | --- | --- |
| | 0 | 1 | 2 | 3 | 4 | 5 |
| $\pi$ | 0.69 | 0.11 | 0.04 | 0.04 | 0.08 | 0.03 |
| $\mu$ | -1.61 | -5.67 | -5.45 | -0.56 | 0.92 | 1.82 |
| [95%CI] | - | [-5.70; -5.64] | [-5.49; -5.41] | [-0.61; -0.51] | [0.89; 0.94] | [1.78; 1.86] |
| $s \times 100$ | 0.00 | 4.46 | 3.56 | -1.47 | -1.92 | -2.10 |
| [95%CI] | - | [4.43; 4.48] | [3.53; 3.59] | [-1.53; -1.41] | [-1.95; -1.89] | [-2.14; -2.07] |

(a)

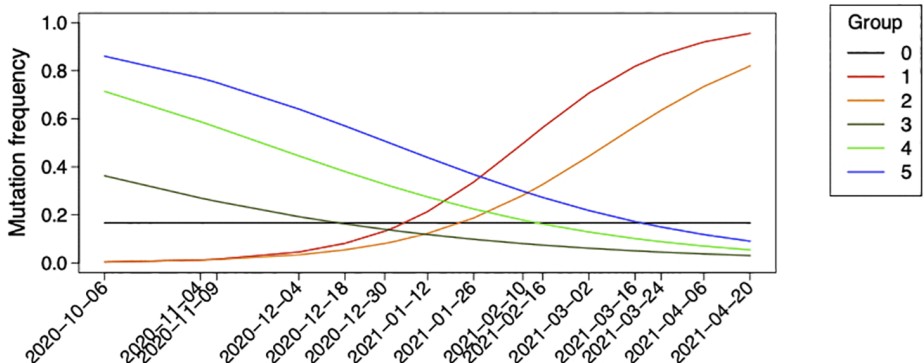

(b)

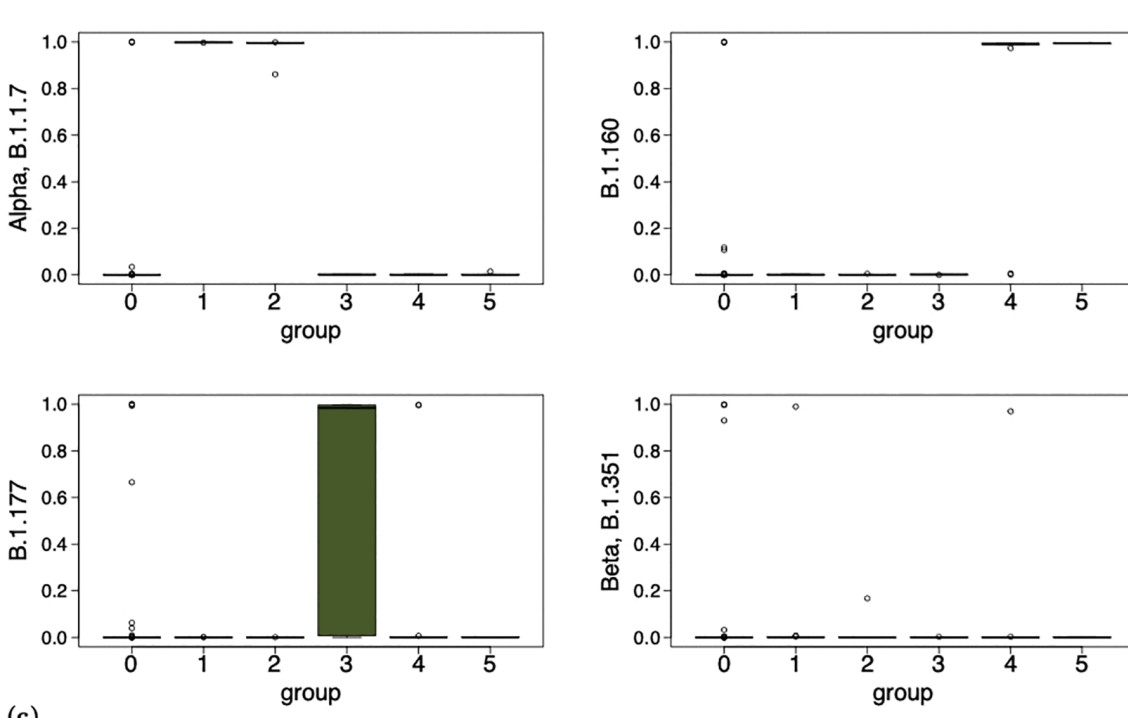

(c)

**Fig 3**. **Analysis of WWTP2 dataset over its entire time period (`WWTP2-2020-Oct-2021-April`).** Parameter estimates (a), associated group frequency trajectories (b) and boxplots of mutation profile stratified on MAP of group assignment (c) computed conditional on WWTP2 dataset over all time points (from 2020-10-06 to 2021-04-20) and $K = 5$ non-neutral groups. Selection coefficients and their 95%CI boundaries are multiplied by 100 in (a). The estimate $\hat{\mu}_0$ in (a) is computed from estimates $\hat{\alpha} = 0.50$ and $\hat{\beta} = 2.51$. Dates 2020-11-08 and 2021-01-02, also composing dataset WWTP2, are removed from x-axis labeling in (b) in order to avoid overlap.

**Table 1**. Mutations composing rising groups in the analyses of WWTP1 and WWTP2 over their entire time period.

| | Unsupervised clustering | | | | | Mutation profile (Virpool) | | |
| | WWTP1 | | | WWTP2 | | | | |
| | $G_1$ | $G_2$ | $G_3$ | $G_1$ | $G_2$ | Alpha | B.1.1 | Beta |
|---|---|---|---|---|---|---|---|---|
| C913T | x | | | x | | 1 | | |
| C3267T | x | | | x | | 1 | | |
| C5388A | x | | | x | | 1 | | |
| C5986T | | | | x | | 1 | | |
| T11296G | | x | | | x | 0.9 | 0.2 | |
| C14676T | x | | | x | | 1 | | |
| C15279T | x | | | x | | 1 | | |
| T16176C | x | | | x | | 1 | | |
| C23271A | | | x | x | | 1 | | |
| C23604A | x | | | x | | 1 | | |
| C23709T | | | x | x | | 1 | | |
| T24506G | | x | | x | | 1 | | |
| G24914C | x | | | | x | 1 | | |
| C27972T | x | | | | x | 1 | | |
| G28048T | x | | | | x | 1 | | |
| A28111G | | | x | x | | 1 | | |
| G28280C | | | x | | x | 1 | | |
| A28281T | x | | | | x | 1 | | |
| T28282A | | | x | | x | 1 | | |
| C28977T | | | | x | | 1 | | |
| G28881A | | | | x | | 1 | 1 | |
| G28882A | x | | | x | | 1 | 1 | |
| G28883C | x | | | x | | 1 | 1 | |
| A23063T | x | | | x | | 1 | | 1 |
| G4136T | | x | | | | | | |
| C2453T | | x | | | | 0.1 | | |
| C6027T | | x | | | | | | |
| C24418T | | x | | | | | | |

List of mutations composing the estimated rising groups in Analyses `WWTP1-2020-Oct-2021-April` (first three columns) and `WWTP2-2020-Oct-2021-April` (middle two columns) where group assignment is materialized with a cross "x". Group numbering and color code are those of Figs 2 and 3 for WWTP1 and WWTP2 respectively. Additionally mutation profiles for main circulating VOCs, computed with Virpool package [40], are provided in the last three columns where probabilities are rounded at one digit after comma and a probability below 0.05 is left empty. Among B.1, B.1.1, B.1.160, Alpha, B.1.177 and B.1.351, only VOCs associated to at least one probability above 0.05 for at least one mutation were reported (that is Alpha, B.1.1 and Beta). For readability, mutations are gathered, in the table, first according to their profile and second according to their position on the genome. Alpha mutations C5986T and C28977T were absent from dataset WWTP1 (after pretreatment with frequency threshold) and shared Alpha & B.1.1 mutation G28881A was assigned to the neutral group in Analysis `WWTP1-2020-Oct-2021-April`.

Similarly we note the absence of any Beta group in both analyses, explained by the fact that all Beta mutations are shared with at least one other main VOC, except C28253T, which is Beta characteristic and present in WWTP1 dataset. This mutation was assigned to the neutral group in Analysis `WWTP1-2020-Oct-2021-April`. However, we notice that decreasing the lower limit of group sizes to 3 mutations for selecting $K$ (see results of this latter analysis in Supporting information S1 Fig) enabled to reveal a fourth rising group $G_4$ composed of 7 mutations among which C28253T (the only Beta characteristic one) as well as T11296G associated with a probability 16.8% (respectively 86.1%) to belong to Beta (respectively Alpha). It is associated with one of the lowest intercept and the lowest selection coefficient among positive ones (table on top of Supporting information S1 Fig) which is consistent with Beta dynamics at the period of time of the analysis, under the assumption of a constant selection coefficient, as Beta emerged later in time. This result reveals in particular the precaution to take when reducing the enormous amount of near zero frequency mutations during dataset preparation. We indeed chose to remove mutations of frequency below 0.05 in a quarter of the samples, which could

be unadapted for revealing a variant of later emergence. According to the question raised, one may apply alternative strategies.

Finally, lowering group size threshold to 3 mutations over WWTP2 dataset (see results of that analysis in Supporting information S2 Fig) highlighted Group $G_5$ in WWTP2 with the highest intercept estimates $\hat{\mu}_5 = 5.00$ hence $\hat{f}_5(0) = 0.99$ and near zero selection coefficient estimate, although still significantly different from zero with a threshold at 2.5% as its 95% CI does not contain value zero. As shown on the graph of mutation profiles stratified on MAP of group assignment in Supporting information S2 Fig, Group $G_5$ is exclusively composed of mutations shared by all main circulating VOC at the time of the analysis. These four mutations (C241T, C3037T, C14408T and A23403G) are precisely the WWTP2 Alpha mutations previously mentioned and assigned to the neutral group when analysing WWTP2 conditional on group size lower limit at 5 mutations. This result echoes the previous comment, a dominant variant being replaced by another one leading to a very high and nearly constant trajectory of mutations shared by all main lineages. Finally, rising and declining groups estimated in previous analyses have roughly been cut into several subgroups.

**Comparison to a posteriori Alpha and B.1.160 VOC frequency through time.** For a better visualization of Alpha and B.1.160 estimated group frequency trajectories, both datasets were analyzed conditional on $K = 2$ non-neutral groups (see Supporting information S3 Fig for a graphical representation of the results of that latter analysis). We obtained, for both datasets, one rising group $G_1$ exclusively composed of 20 (WWTP1) and 23 (WWTP2) Alpha mutations and one declining group $G_2$ exclusively composed of 12 B.1.160 mutations for WWTP1 and exclusively composed of 16 B.1.160 plus 2 B.1.177 mutations in WWTP2. One to two mutations per group are shared either with B.1.1 or Beta. Parameters are respectively estimated at $\hat{\mu}_1 = -3.49$, $\hat{\mu}_2 = 1.09$, $\hat{s}_1 = 3.02 \times 10^{-2}$, $\hat{s}_2 = -2.05 \times 10^{-2}$ for WWTP1 and $\hat{\mu}_1 = -5.70$, $\hat{\mu}_2 = 1.19$, $\hat{s}_1 = 4.20 \times 10^{-2}$, $\hat{s}_2 = -1.96 \times 10^{-2}$ for WWTP2. We reported Alpha and B.1.160 VOC frequencies provided by Nextstrain along with our Group $G_1$ and Group $G_2$ estimated frequencies at a selection of time points in Table 2. The selection of time points is simply motivated by a subset of those chosen by Nextstrain.

We can see that our model captures the global tendency of Alpha but tends to overestimate its frequency until mid-November 2020, underestimate it between mid-November 2020 and the end of January 2021 and overestimate it from February 2021 until the end of March 2021. On the contrary B.1.160 frequencies are underestimated by our model until mid-November and overestimated afterwards until the beginning of March 2021. These alternations between overestimation and underestimation are explained, as previously mentioned, by the fact that we assume a constant selection coefficient over the time period studied and therefore, a global estimate, with no consideration of time-varying

**Table 2**. Estimated group frequencies versus Alpha and B.1.160 VOC reported by Nextstrain.

|  | 23/10 2020 | 07/11 2020 | 25/11 2020 | 24/12 2020 | 22/01 2021 | 24/02 2021 | 25/03 2021 |
|---|---|---|---|---|---|---|---|
| Alpha (Nextstrain) | <1 | <1 | 11 | 43 | 39 | 47 | 59 |
| Group ↗ WWTP1 | 3 | 5 | 8 | 18 | 34 | 59 | 77 |
| Group ↗ WWTP2 | 1 | 1 | 3 | 9 | 24 | 56 | 81 |

Alpha & emerging group in datasets conditional on $K = 2$

|  | 23/10 2020 | 07/11 2020 | 25/11 2020 | 24/12 2020 | 22/01 2021 | 24/02 2021 | 25/03 2021 |
|---|---|---|---|---|---|---|---|
| B.1 (Nextstrain) | 84 | 70 | 47 | 38 | 19 | 8 | 11 |
| Group ↘ WWTP1 | 74 | 67 | 59 | 44 | 30 | 18 | 11 |
| Group ↘ WWTP2 | 76 | 69 | 60 | 43 | 28 | 15 | 9 |

B.1.160 & declining group in datasets conditional on $K = 2$

Alpha (top table) and B.1.160 (bottom table) VOC frequency in France reported by Nextstrain (first row of each table) at various time points along with Group $G_1$ (rising) and Group $G_2$ (declining) frequencies estimated through the analysis of WWTP1 and WWTP2 datasets over their respective entire time period, between October 2020 and April 2021, and conditional on $K = 2$ non-neutral groups. Frequencies are given in percent.

changes influenced by vaccinations, the late emergence of a new variant, etc. In order to overcome this limitation, we plan on including time dependent (starting with piecewise constant) selection coefficients and perform break point detection in the future.

**Comparison with supervised results.** Let us return to Figs 2 and 3 along with Table 1 for comparing our unsupervised results to supervised ones of Barbe et al. (2022) in [36]. Groups $G_1$ and $G_3$ in Analysis `WWTP1-2020-Oct-2021-April` are both of positive selection coefficient estimates (Fig 2a and 2b) and exclusively composed of Alpha mutations (Fig 2c and Table 1). Their intercept are clearly distinct with frequency at time origin $\hat{f}_3(0) = 0.058 > \hat{f}_1(0) = 0.014$. This results suggests that Group $G_3$ mutations appeared in WWTP1 earlier than those of $G_1$. It is in line with Figure 5 in [36] as all $G_3$ (respectively $G_1$) mutations are detected on 2020-11-17 or earlier (respectively after 2020-11-17) by Barbé et al. except C3267T and A28281T which are detected respectively on 2020-10-20 and 2020-11-17 by the authors and assigned to Group $G_1$ by our model. Note however that Figure 5 of the authors reveals the absence of detection of C3267T between 2020-11-17 and 2021-02-02 and a rapid increase from 2021-02-02 which could explain its assignment to $G_1$ of higher selection coefficient ($\hat{s}_1 > \hat{s}_3$, Fig 2a).

Similarly, Groups $G_1$ and $G_2$ in Analysis `WWTP2-2020-Oct-2021-April` are both of positive selection coefficient estimates (Fig 3a and 3b) and exclusively composed of Alpha mutations (Fig 3c and Table 1). They are associated to similar intercept estimate with frequency at time origine $\hat{f}_1(0) = 0.003$ and $\hat{f}_2(0) = 0.004$. The steeper increased frequency of Group $G_1$ ($\hat{s}_1 = 4.46[4.43; 4.48] > \hat{s}_2 = 3.56[3.53; 3.59]$) (visually) appears to be in line with Figure 5 in Barbé et al. except for C913T assigned to $G_1$ although it does not seem to be of overall steep increased frequency in [36]. The assignment of C913T to $G_1$ could be explained by the absence of its detection from 2020-12-30 until 2021-02-10 reported by the authors leading to a rapid increase around February 2021.

In brief, both analyses `WWTP1-2020-Oct-2021-April` and `WWTP2-2020-Oct-2021-April` show the adequacy between our results and those of Barbé et al. with the noticeable difference that, since unsupervised, our method does not require any prior knowledge on the status of mutations. Moreover we provide parameter estimates such that one can quantify the strength of the selection.

### Fitness detection

In order to assess the performances of our model in detecting new variants of increased fitness early in time, we performed several analyses over WWTP1 and WWTP2 datasets restricted to the time period of Alpha emergence, between the end of October and mid-December 2020. Each analysis will be denoted with the dataset, the year of the analysis, the starting month and day, the ending month and day, separated with sign "-". In Analysis `WWTP1-2020-10-20-11-04` (respectively `WWTP1-2020-11-04-11-17`), WWTP1 dataset is restricted to the first two time points (2020-10-20 and 2020-11-04) (respectively the second and third time points, 2020-11-04 and 2020-11-17). In Analysis `WWTP2-2020-11-09-12-04` (respectively `WWTP2-2020-12-04-12-18`), WWTP2 dataset is restricted to time points 2020-11-09 and 2020-12-04 (respectively 2020-12-04 and 2020-12-18). The choice of time origin for WWTP2 is motivated by Figure 6 in [36] as Alpha mutations were detected above 5% from 2020-12-18 in WWTP2 and a month earlier (2020-11-17) in WWTP1. For each dataset, mutations of frequency below 0.05 in both samples were removed. Moreover and as mentioned in previous analyses, a read depth below 10 is set to zero (along with related mutation counts). An additional dataset reduction was however needed in order to reduce the extra noise induced by the restriction to two time points and gain in statistical power (see Supporting information S5 Text for an illustration of this issue). We propose to remove mutations associated with a probability strictly below 0.005 to belong to B.1, B.1.1 and/or B.1.160, that is main VOC known before the studied time period. In such manner we do not use any information regarding unknown VOC (in particular Alpha) at the beginning of the time period studied (October/November 2020). Moreover, with a threshold as low as 0.005, we keep most mutations of interest and we may remove part of sequencing errors and transient variants. This strategy should however be adapted to the context, datasets and time periods of interest and further investigations over a variety

of datasets are in progress for future works to refine the process of dataset reduction. We finally obtained datasets composed of 33, 37, 30 and 34 mutations respectively for Analyses `WWTP1-2020-10-20-11-04`, `WWTP1-2020-11-04-11-17`, `WWTP2-2020-11-09-12-04` and `WWTP2-2020-12-04-12-18`.

The BIC, ICL and $-2 \log L$ of models composed of $K = 1$ to $K = 10$ groups are pictured in Fig 4 for each analysis, along with the size of the smallest group, for each tested $K$, on the top axis. We faced the same issue as in previous analyses for selecting the number of non-neutral groups $K$, with a BIC and ICL decreasing with an increasing number $K$ and no clear elbow in the log-likelihood. In such context, we refer to the same alternative option based on the size of the smallest group with a lower limit at 3 mutations for a dataset covering a time period in weeks leading to a choice of $K = 1$, $K = 2$,

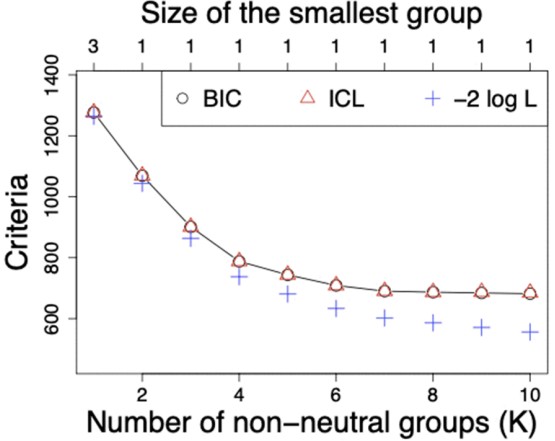

WWTP1 from 2020-10-20 to 2020-11-04

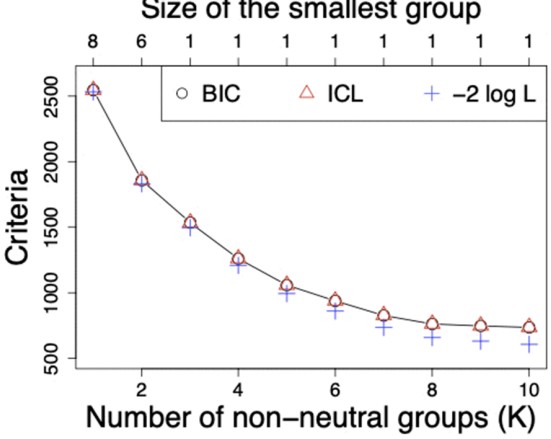

WWTP1 from 2020-11-04 to 2020-11-17

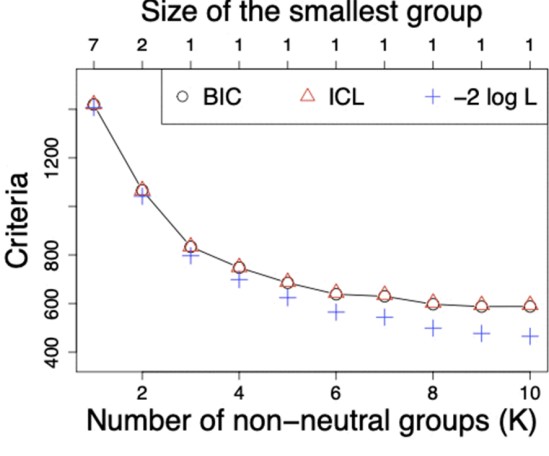

WWTP2 from 2020-11-09 to 2020-12-04

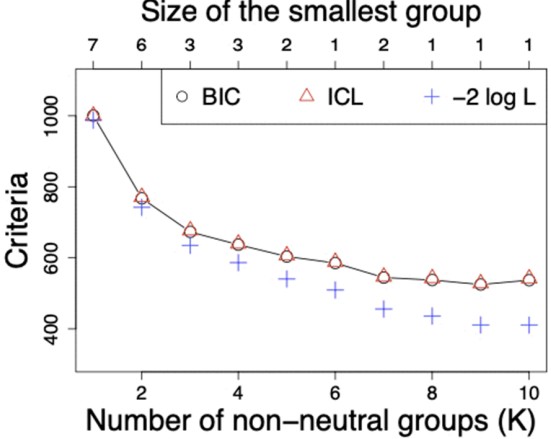

WWTP2 from 2020-12-04 to 2020-12-18

**Fig 4**. **Selection of the number of groups in WWTP1 and WWTP2 datasets restricted to time periods between October and December 2021.** BIC, ICL along with minus two times the log-likelihood ($-2 \log L$) of models composed of $K = 1$ to $K = 10$ non-neutral groups with WWTP1 dataset restricted to time points 2020-10-20 & 2020-11-04 (top left) as well as 2020-11-04 & 2020-11-17 (top right) and with WWTP2 dataset restricted to time points 2020-11-09 & 2020-12-04 (bottom left) as well as 2020-12-04 & 2020-12-18 (bottom right). Each dataset is also reduced to mutations associated with a probability above 0.005 to belong to B.1, B.1.1 and/or B.1.160. The size of the smallest group (in terms of number of mutations) is reported on the top axis.

$K = 1$ and $K = 4$ non-neutral groups respectively for Analyses `WWTP1-2020-10-20-11-04`, `WWTP1-2020-11-04-11-17`, `WWTP2-2020-11-09-12-04` and `WWTP2-2020-12-04-12-18`.

Estimated group frequency trajectories along with their profile for Alpha and B.1.160 VOC stratified on MAP of group assignment are given in Figs 5 and 6 for WWTP1 and WWTP2 respectively. Absolute values of estimated selection coefficients are about ten times higher than those estimated over the entire time period, until April 2021, which is consistent with the time period of data collection, before and during the second lockdown in France. We can notice that all non-neutral mutations in Analyses `WWTP1-2020-10-20-11-04` and `WWTP2-2020-11-09-12-04` are associated to a probability above 0.98 to belong to B.1.160 which is consistent with the fact that B.1.160 was dominating before its replacement by Alpha. No Alpha mutation was detected before 2020-11-04 (respectively 2020-12-04) in WWTP1

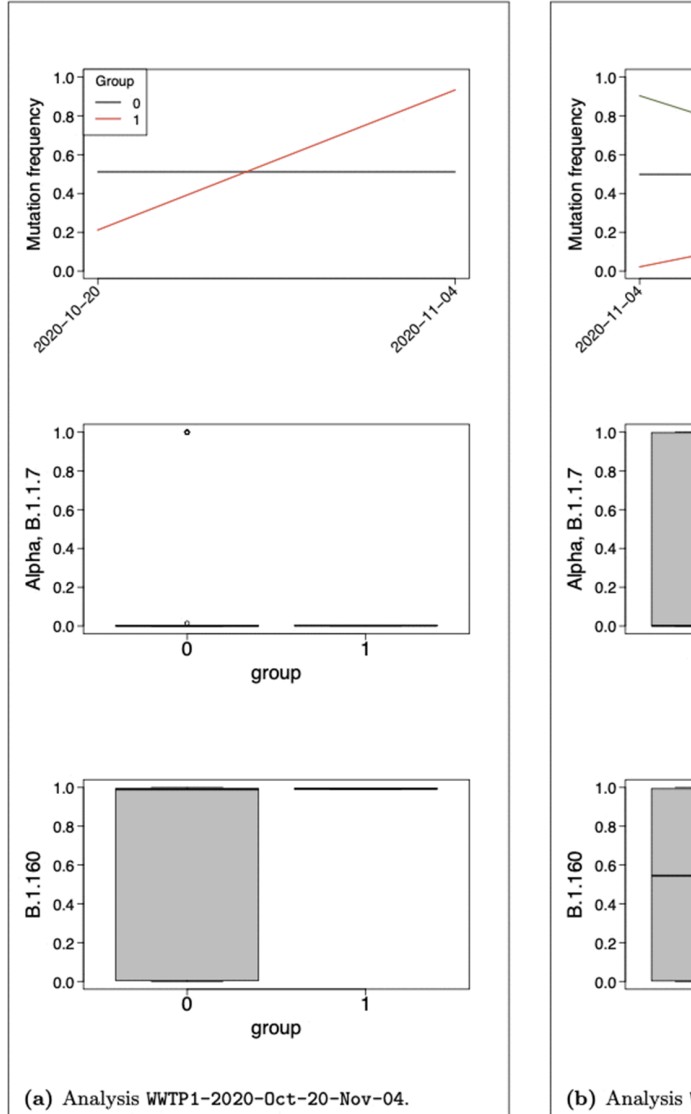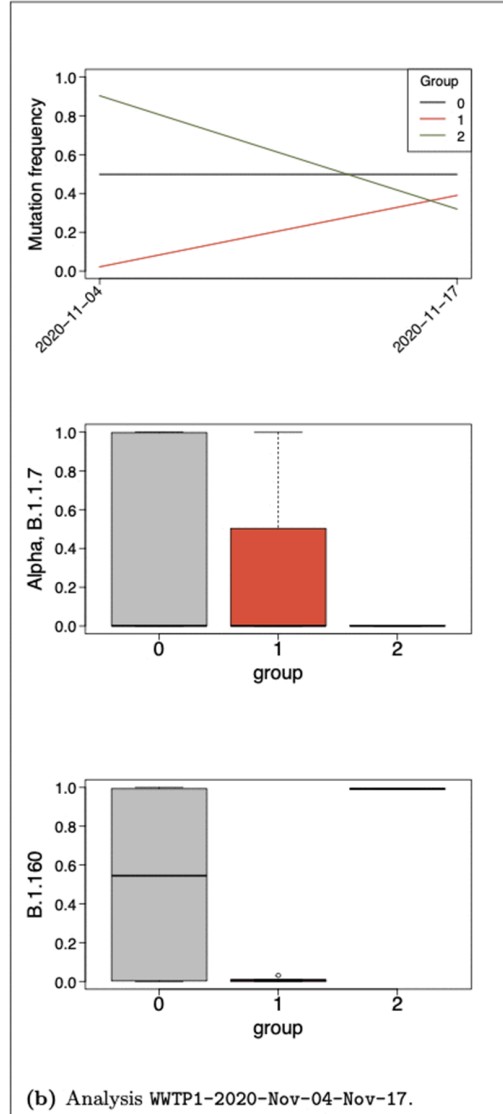

**Fig 5**. **Analyses of WWTP1 dataset over restricted time periods between October and November 2021.** Each panel is associated to one analysis, one period of time. Group frequency trajectories (top) along with boxplots of mutation profile for B.1.1.7 (middle) and B.1.160 (bottom) VOC stratified on MAP of group assignment. The number of non-neutral groups is driven by a lower group size limit at 3 mutations.

**Fig 6. Analyses of WWTP2 dataset over restricted time periods between November and December 2021.** Each panel is associated to one analysis, one period of time. Group frequency trajectories (top) along with boxplots of mutation profile for B.1.1.7 (middle) and B.1.160 (bottom) VOC stratified on MAP of group assignment. The number of non-neutral groups is driven by a lower group size limit at 3 mutations.

(respectively WWTP2), as Barbé et al. showed in their Figure 6 with a threshold at 5%. Declining groups in the analyses of subsequent time periods are entirely composed of B.1.160 mutations for both datasets except Group $G_4$ in Analysis `WWTP2-2020-12-04-12-18` which contains four B.1.160 mutations and G8102T not associated to any circulating VOC. Additionally, `WWTP1-2020-11-04-11-17` (respectively `WWTP2-2020-12-04-12-18`) reveals one (respectively two) rising group(s) partly composed of Alpha mutations.

The set of mutations composing rising groups along with their profile for main VOC at the time of the analyses are listed in Table 3. Consistently, the two Alpha mutations detected in Analysis `WWTP1-2020-11-04-11-17`, C3267T and A28111G, are two of the seven Alpha mutations detected by Barbé et al. on 2020-11-17 in WWTP1 (see Figure 5 in [36]). Similarly, Alpha mutations assigned Group $G_1$ in Analysis `WWTP2-2020-12-04-12-18` (C14676T and A28111G) belong

Table 3. Mutations composing rising groups in the analyses of WWTP1 and WWTP2 over restricted time periods.

| | Unsupervised clustering | | | Mutation profile (Virpool) | | | |
|---|---|---|---|---|---|---|---|
| | **WWTP1** | **WWTP2** | | | | | |
| | $G_1$ | $G_1$ | $G_2$ | **Alpha** | **B.1** | **B.1.1** | **B.1.160** |
| C913T | | | x | 1 | | | |
| C3267T | x | | | 1 | | | |
| C14676T | | x | | 1 | | | |
| C15279T | | | x | 1 | | | |
| A28111G | x | x | | 1 | | | |
| G28882A | | x | | 1 | | 1 | |
| G28883C | | x | | 1 | | 1 | |
| G25563T | | | x | | 0.56 | | 0.99 |
| G571A | x | | | | | | |
| C1059T | | | x | | 0.45 | | |
| T5071C | x | | | | | | |
| C11916T | x | | | | | | |
| C22227T | x | | | | | | |
| T24982C | | | x | | | | |
| T27212C | | | x | | | | |
| T27384C | x | | | | | | |
| C27945A | | | x | | | | |
| G28086T | | | x | | | | |

List of mutations composing the estimated rising group(s) in Analyses `WWTP1-2020-Nov-04-Nov-17` (first column) and `WWTP2-2020-Dec-04-Dec-18` (second and third columns) where the assignment to a group is materialized with a cross "x". Group numbering and color code are those of Figs 5 and 6 for the related analyses. Additionally the VOC profile of each mutation for Alpha, B.1, B.1.1 and B.1.160, computed with Virpool package, is provided in the last four columns where probabilities are rounded at one digit after comma and a probability below 0.05 is left empty. For readability, mutations are gathered, in the table, firstly according to their profile and secondly according to their position on the genome. Alpha mutation C913T was absent from dataset WWTP1. Shared Alpha & B.1.1 mutations G28882A and G28883C as well as B.1.160 mutation G25563T were assigned to the neutral group in Analysis `WWTP1-2020-Nov-04-Nov-17`.

to those detected by the authors on 2020-12-18 in WWTP2. The shared mutations G28882A and G28883C were reported undetected by the authors on 2020-12-18 in WWTP2 although they were assigned to Group $G_1$ by our model. Their raw frequency (computed from raw count data) indeed increased from 0.00 on 2020-12-04 to 0.45 on 2020-12-18 for both mutations in WWTP2. We also distinguish a rising group $G_2$ in Analysis `WWTP2-2020-12-04-12-18` containing Alpha mutations C913T and C15279T although they were reported as undetected by the authors on 2020-12-18 in WWTP2. This observation seems to be a consequence of applied thresholds as they were counted four or less than four times in both samples and we applied no threshold over mutation count, solely over frequency and read depths.

Let us finally say a word about the set of Alpha mutations assigned to the neutral group in analyses `WWTP1-2020-10-20-11-04` and `WWTP2-2020-12-04-12-18` (C241T, C3037T, C14408T, C27972T, G28881A, G28882A and G28883C). As expected and similarly to the analyses over the entire time period, all these mutations, except C27972T, are shared between Alpha and at least, one other main circulating VOC among B.1, B.1.1, B.1.60 as shared mutations tend to display near constant high frequency trajectories.

These results show the capacity of our model to detect a potentially threatening group of mutations as early as Barbé et al., that is 2020-11-17 for WWTP1 and 2020-12-18 for WWTP2 as shown in their Figure 6, with the noticeable difference that, since unsupervised, our method does not need any prior information on the set of mutations in the dataset while running the algorithm. Although prior knowledge on circulating VOC known before the start date of the analysis (B.1, B.1.1 and B.1.160) was required, in this example, during the process of data preparation for reducing the noise of data, no information on unknown lineages was used. Alternative strategies for dataset reduction may also be applied and tested in future work.

We can finally check that lowering group size limit at 2 mutations leads to equal results for WWTP1 with same chosen $K$ and similar results for WWTP2 as shown in Supporting information S4 Fig. Non-neutral groups over time period 2020-11-09 to 2020-12-04 gather exclusively B.1.160 mutations. Over the subsequent time period, groups of low (respectively high) intercepts and positive (respectively negative) selection coefficient gather partly or exclusively Alpha (respectively B.1.160) mutations. We note an additional group of high intercept and positive selection coefficient which is exclusively composed of two B.1.160 mutations, G22992A and C25710T and a group of nearly constant at 0.99 frequency shared by all main lineages.

## Discussion and perspectives

We presented an unsupervised method for clustering mutation frequency trajectories and estimating group fitness from time series of SARS-CoV-2 genome sequences. Our method takes time series of mutation count data and associated read depths as input and returns an estimated number of groups, group proportions, frequency at time origin and selection coefficient estimates associated to each group as well as group assignment per mutation. Our method is suited for fragmented and pooled genomes of multiple lineage origin, typically found in WW samples. Although only tested on the difficult case of WW sample analysis, it could also be applied to an aggregation of clinical samples.

We applied our method to publicly available WWTP datasets presented in [36] and collected between October 2020 and April 2021. We demonstrated that our method highlights groups of mutations whose frequency trajectory estimates and in particular frequency at time origin and selection coefficient estimates are consistent with the observations of the authors of [36] regarding Alpha and B.1.160 VOC as well as VOC dynamics presented in Nextstrain reports for Alpha, B.1.160 and B.1.177 as well as, to a lower extent, Beta. We showed its capacity to group Alpha mutations into distinct groups according to their temporal emergence consistently with the time of their detection by Barbé et al. in WWTP1 or according to the steepness of their increased trajectory in WWTP2. Restricting the analysis to a period of time shortly before and at the beginning of the emergence of Alpha (until 2020-11-17 for WWTP1 and 2020-12-18 for WWTP2) leads to detect groups of high positive selection coefficient exclusively or partly composed of Alpha mutations detected by Barbé et al. [36] at the same date. Moreover, our parsimonious model fits in limited time complexity, all aforementioned analyses being performed in less than 6 minutes (less than ten seconds for datasets reduced to two time points) when applying parallel computation. In summary, our results are consistent with those of Barbé et al. and estimated variant frequencies in France reported by Nextstrain over the time period considered with the noticeable difference that our method is unsupervised, that is, it does not require any prior knowledge on the set of mutations contained in the dataset. It is therefore suited for detecting newly emerging variants.

Information on lineages known before the analysis was however used during the process of data preparation when analyzing short time periods. Preparing datasets with a reduction step is needed for limiting the enormous amount of near zero frequency mutations mostly due to sequencing errors and transient variants (recent mutations under purifying selection or in the process of stochastic extinction). Our choice of applying a threshold at 0.005 on the probability to belong to a previously known lineage (B.1, B.1.1 and/or B.1.160) proved, a posteriori, to be a commendable choice for the unsupervised detection of Alpha over weekly time periods at the end of 2020. However, it would probably be inadequate for an analysis at the beginning of a pandemic as one future emerging variant may have no mutation in common with circulating dominant lineages at that time, even with a threshold as low as 0.005. For analyses covering longer time periods (from October 2020 to April 2021), we chose to apply a threshold at 0.05 for frequencies in a quarter of the samples with no consideration on any variant. This strategy proved to be valuable to distinguish groups of Alpha, B.1.160 and B.1.177 mutations but washed away almost all Beta characteristic mutations from the dataset. The need of dataset reduction is one of the main drawbacks of the method and still needs to be empirically clarified, thresholds adapted and/or alternative options tested over various datasets. As an example of alternative reduction strategy, one could choose to focus on genomic regions of interest, for instance those related to the Spike protein.

Our method presents some other limitations. Its main weakness is the lack of robust criteria, such as Bayesian criteria, for determining the number of groups to select. This task was however well performed over simulated datasets. Nor was any distinguishable elbow in the log-likelihood of models composed of an increasing number of non-neutral groups. We proposed an alternative option based on the size of the smallest group which provides satisfying results over various datasets but it still needs to be empirically tested in broader frameworks. The absence of a clear and robust method for determining the number of groups may be structurally inherent to the model as one group is not associated to one lineage stricto sensu but it is a set of mutations with similar frequency trajectory. Shared mutations among several lineages and/or the temporal emergence of mutations through time may induce an accumulation of subgroups with their own parameter estimates.

In order to lift the aforementioned limitations, we intend, in future work, to pursue empirical investigations over a variety of WW datasets and clinical samples in order to design an optimal methodology for dataset reduction, number of groups selection as well as a threshold for selection coefficients in various context (various time periods, dataset sizes, sequencing technologies, etc.). Moreover we plan to enrich that future article with more theoretical considerations, that goes beyond the current applied work, for instance regarding the selection of the number of groups or algorithm time complexity. Moreover, we did not encounter any parameter identifiability issue across extensive, various and targeted simulation studies and we would like to study the potential correlation among parameters and parameter identifiability in a theoretical way in this future work.

The lack of criteria for the selection of the number of groups also reflects the poor capacity of our method in capturing residual variability to which SARS-CoV-2 variants dynamics, with multiple circulating lineages, and WWTP data are particularly prone. Along with the variance inherent to multinomial and binomial distributions, the only additional variability lays in the beta-binomial distribution of neutral mutations counts. We indeed assumed a generalized linear model with a binomial family with fixed effects (constant parameters) for modeling the distribution of mutation counts conditional on a non-neutral group assignment. One could relax such assumption with a mixed effect model composed of a random intercept and/or random selection coefficient for non-neutral groups, for instance Gaussian parameters. However, a drastic increased time complexity would be inevitable when marginalizing over random effects.

We also assumed constant selection coefficients restricting our method to limited periods of time, which can be sequentially repeated. That limitation leads to ignoring break points in frequency trajectories which characterizes the emergence of a mutation conferring a selective advantage. We therefore would like to further develop the model with time dependent selection coefficients and break point detection.

In order to overcome these latter two limitations, our second main perspective is the development of the hidden random walk model described in Supporting information S3 Text in order to better capture the structure dependency of our variables through time and add variability. We intend to pursue its development with time-dependent selection coefficients in a first place and break point detection in a second place. Such model is however computationally much more intensive.

Note finally that WWTP datasets are highly fragmented but still partly contain haplotype information that would be a valuable input to be taken into account in future developments.

## Supporting information

**S1 Text. Statistical method.**
(PDF)

**S2 Text. Model assessment over simulated datasets.**
(PDF)

**S3 Text. The hidden random walk model.**
(PDF)

**S4 Text. Model entropy.**
(PDF)

**S5 Text. Additional dataset reduction for analyses covering short time periods.**
(PDF)

**S1 Fig. Analysis of WWTP1 conditional on group size lower limit at three mutations.**
(PDF)

**S2 Fig. Analysis of WWTP2 conditional on group size lower limit at three mutations.**
(PDF)

**S3 Fig. Analyses conditional on $K = 2$ non-neutral groups.**
(PDF)

**S4 Fig. Analysis of WWTP2 at the beginning of Alpha emergence conditional on group size lower limit at two mutations.**
(PDF)

## Acknowledgments

This work has benefited from the environment of the multidisciplinary scientific group of interest GIS-OBEPINE (Isabelle Bertrand, Mickaël Boni, Christophe Gantzer, Soizick Le Guyader, Yvon Maday, Vincent Maréchal, Jean-Marie Mouchel, Laurent Moulin). We are very thankful to Laure Barbé and Marion Desdouits for their help in data collection, preprocess and for providing us with aligned bam files. We warmly thank Marie Courbariaux, Nicolas Cluzel and El Hacene Djaout for their help in data preparation and their feedbacks on statistical methods. We deeply thank Stéphane Robin and Grégory Nuel for their generous advice and for sharing their knowledge in computational statistics. We also would like to thank Sébastien Wurtzer and Philippe Lopez for their explanations regarding data preparation and variant calling as well as Samuel Alizon for his precious advice and discussions on viral evolution and epidemiology.

## Author contributions

**Conceptualization:** Alexandra Lefebvre, Vincent Maréchal, The Obépine Consortium, Amaury Lambert, Yvon Maday.

**Data curation:** Alexandra Lefebvre, Arnaud Gloaguen.

**Formal analysis:** Alexandra Lefebvre, Yvon Maday.

**Funding acquisition:** Vincent Maréchal, The Obépine Consortium, Yvon Maday.

**Methodology:** Alexandra Lefebvre, Amaury Lambert, Yvon Maday.

**Project administration:** Vincent Maréchal, Yvon Maday.

**Software:** Alexandra Lefebvre.

**Supervision:** Vincent Maréchal, Amaury Lambert, Yvon Maday.

**Visualization:** Alexandra Lefebvre.

**Writing – original draft:** Alexandra Lefebvre.

**Writing – review & editing:** Alexandra Lefebvre, Vincent Maréchal, Arnaud Gloaguen, Amaury Lambert, Yvon Maday.

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
