## [Decision Letter · Decision Letter 0]

6 Apr 2025

PCOMPBIOL-D-25-00088

Unsupervised detection and fitness estimation of emerging SARS-CoV-2 variants: Application to wastewater samples (ANRS0160)

PLOS Computational Biology

Dear Dr. Lefebvre,

Thank you for submitting your manuscript to PLOS Computational Biology. After careful consideration, we feel that it has merit but does not fully meet PLOS Computational Biology's publication criteria as it currently stands. Therefore, we invite you to submit a revised version of the manuscript that addresses the points raised during the review process.

Please submit your revised manuscript within 60 days Jun 06 2025 11:59PM. If you will need more time than this to complete your revisions, please reply to this message or contact the journal office at ploscompbiol@plos.org. Please include the following items when submitting your revised manuscript:

We look forward to receiving your revised manuscript.

Kind regards,

Andrey Rzhetsky

Academic Editor

PLOS Computational Biology

Roger Kouyos

Section Editor

PLOS Computational Biology

**Journal Requirements:**

At this stage, the following Authors/Authors require contributions: Alexandra Lefebvre, Vincent Maréchal, Obépine Consortium, Arnaud Gloaguen, Amaury Lambert, and Yvon Maday. Please ensure that the full contributions of each author are acknowledged in the "Add/Edit/Remove Authors" section of our submission form.

4) We noticed that you used the phrase 'not shown' in the manuscript. We do not allow these references, as the PLOS data access policy requires that all data be either published with the manuscript or made available in a publicly accessible database. Please amend the supplementary material to include the referenced data or remove the references.

5) Please upload all main figures as separate Figure files in .tif or .eps format. For more information about how to convert and format your figure files please see our guidelines: 

6) Please ensure that all Figure files have corresponding citations and legends within the manuscript. Currently, Figures 3, and 4 in your submission file inventory do not have in-text citations. Please include the in-text citations of the figures.

7) Thank you for stating "Code for data processing is available at https://github.com/AlexandraLefe." Please provide us with specific link to access the code.

8) Please amend your detailed Financial Disclosure statement. This is published with the article. It must therefore be completed in full sentences and contain the exact wording you wish to be published.

Please ensure that the funders and grant numbers match between the Financial Disclosure field and the Funding Information tab in your submission form. Note that the funders must be provided in the same order in both places as well.

**Reviewers' comments:**

Reviewer's Responses to Questions

Reviewer #1: Dear authors,

Thank you for the opportunity to review the manuscript titled ‘Unsupervised detection and fitness estimation of emerging SARS-CoV-2 variants: Application to wastewater samples (ANRS0160)’. I am overall very impressed with this paper, it is clearly a large quantity of high standard work, and presented with thorough technical details. I have no problems at all with the analyses, outcomes, and general conclusions, I think these aspects are acceptable for publication as is. However, I believe the presentation of the manuscript needs strong improvements in Methods and Results sections. The principal problem is that the Results are too long, too detailed, and contains too much content that should have gone into the Methods section. While the overall narrative is clear and cohesive, it takes the reader much effort to step into the results to specifically look for results that support your discussions and conclusions. I suggest the authors make liberal use of space of appendices, and move much of the result texts, figures, and tables out of the main text, since much of it is repeated across different analyses. I believe these changes would really strengthen the paper, and make it much easier to read for readers unfamiliar with the complicated nature of latent mixture models. With these substantial but straightforward editorial changes, I believe the manuscript will be ready for publication.

Please find below more specific comments:

Line 34 – 35: variant classifications are an outcome of the variant’s biological characteristics (transmissibility, virulence, immune evasion etc). I think in the first paragraph you should explain why some variants are of concern (e.g., more transmissible, more severe outcomes, or overcoming existing immunity)

Line 41: interest in wastewater genomic surveillance specifically? There is strong interest in using wastewater as an epidemic trend indicator. I think you need to clarify the context of using WW for genomic surveillance of viral evolution, and therefore it is complementing clinical samples (just so readers don’t confuse clinical samples as just PCR positivity)

Line 47: specify here the clinical samples you talk about are individual-specific, in some regions and some phases of the pandemic pooled clinical sampling are used

Line 55: why are they inefficient? Is it because we lack information on what specific mutations encode threatening phenotypes? It’s ambiguous why these approaches are inefficient.

Section 2.1: I would like to see a description of the model before the data. In introduction you said you would give general introduction of the model, and apply it to simulations, then finally to real-life data. It is confusing then to see the data presented first.

Fig 1: the date labels on x axes are not clear – there are overlaps and they are generally small. I suggest stretching out both panels to be wider, and stack them top and bottom, to help visualise the change over time. Also consider adding date ticks. It is a bit difficult to appreciate how many days are in between sampling as you have to mentally calculate rather than read off the graph.

Lines 130-131: is K = 0 the neutral group or non-neutral? I am confused, the text here seems to imply it is non-neutral but later on you describe it as if it is the neutral group.

Line 136: note the line number formatting changed in this section. The X_{i,t} (respectively d_{i,t}) sentence is confusing – I got it on second read but I think you should just describe the two quantities in separate sentences.

Top of page 8: The group specific intercepts mu_k are the baseline prevalence of group k at the start of the time series is that right? I think you should explicitly state it to remind readers your groups don’t start at the same level (i.e., the observed prevalence is an outcome of the initial condition and relative fitness)

Please number your equations for review purposes.

First equation on page 8: you say the selection coefficient makes no distinction between evolutionary or epidemiological parameters. This implicitly assumes the time-varying force of infection contributes to the rate of variant replacement. However your neutral group model has no time-varying component at all – would you expect time-varying FOI to have an effect on random mutation responses too? E.g., if a random mutation happens to have happened as a major epidemic wave begins to rise (but it does not confer an advantage), would you still expect its relative frequency to be still consistent with a time constant beta binomial? I don’t have an answer here either but I suspect it might drive your beta-binomial to have a strong degree of overdispersion.

One line below line 137: wouldn’t you have just K pi parameters instead of (K+1) since sum of pi is 1, so you get a free pi_0?

Line 139: it might be easier to define the binomial rate parameter with a new symbol rather than writing out the logit every time.

Line 144: my experience is that initialisation problem happens with EM/ECM on almost all real data sets. I would frame this paragraph as general approach to the method, not just specifically for the dataset. Related to this, one might also start with quasi-Newton or another faster and less precise method and swap to EM if you are stuck in a flat likelihood region upon initialisation.

More generally, does the initialisation trick help with label-switching? I think you need to flag label switching as a common issue for these approaches. You did discuss label switching later in the context of validating against simulated data, but at that point an unfamiliar reader won’t know what label switching means.

Line 163: simulations of synthetic data should be presented in Methods section – it is not obvious why you start results with simulations, or what purpose they serve

Line 169: this paragraph is narratively awkward – you have not yet introduced the general problem of selecting the numbers of groups, so the simulation reads like a self-contained side exercise rather than a necessary interrogation in the EM latent clustering approach. Again I think you need to move some of the content to Methods to smooth out the narrative. It is also not clear to me why and how you decided on the scheme classifications, since within schemes you have variations anyway. I think you need to start the paragraph by highlighting what model behaviour/biological reality you are trying to tease out with each scheme, then describe their associated parameters. I think this should be in Methods section, with results limited to just how well groups are identified from simulated data.

Table 2: the results raise a question: does using beta-binomial for neutral group bias group selection towards neutrality, since it would yield different likelihoods than the binomial non-neutral model? I think in principal it makes sense to treat neutral model as beta-binomial but this should be discussed.

Line 197: the paragraph on parameter estimation is too verbose and again suffers from being a mix of methods and results

Figures 2-4: the legends, axes ticks and labels should all be bigger. They are very hard to read in their current form, especially for a complex multi panel figure. Also I suggest using a different style, symbol, background colour etc to differentiate AUC from the rest, since AUC is a performance metric not a parameter so we want to clearly communicate that to readers.

Line 241: how did you come up with the upper limit for K? From experience I know there is a computational ceiling and higher K would be biologically unreasonable, but you should make it explicit for readers unfamiliar with latent group clustering.

Line 246: have you introduced entropy in Methods?

Figure 5: as expected, BIC and ICL are strongly driven by log L (since parameter space is high and latent mixture models tend to give large absolute values for log L), which raises doubts as to if these criteria are penalising model complexity sufficiently. You can see both criteria appearing to monotonically decrease within your K ranges. This is not reassuring for readers who do not know the caveats about latent mixture models. I feel you need more nuanced text in Methods section to prepare readers for what appears to be poor K selection results at a glance.

Line 258-267: I see you are addressing some of my earlier comments here – these should be moved to Discussion section.

Line 268: you already said you were using elbow method in Methods, the justification should be done there.

Line 296: again, the text for Methods and Results are blending into one another

Figures 6 to 8: I find the inconsistent labelling between group number and “pi-group number” confusing. In general many of the graphical elements can be tweaked to improve presentation, as suggested in my earlier comments.

Generally, I find labelling analyses with letters confusing. You already have many parameters and model structures for readers to keep track of. I think the analyses should be labelled with more intuitive, if more verbose, names, e.g., WWTP1-20-10-21-04 etc. Better yet, find a way to condense a summary narrative out of the various analyses, or consider only showing a few in the main text and moving the rest to appendices.

Overall, the Results section is too verbose. Your average readers will not be interested in all of the different analyses results across time periods, I think much is needed in terms of condensation to strengthen the overall narrative.

Line 615: if the elbow approach worked out well I would say criteria are not the most key limitation. More importantly you could suggest future work to test the robustness of elbow method in K selection. It goes against intuitive to forsake criteria, but if we practically find it working every time then we have confidence in using it.

Line 622: how would you parameterise random effect? I’m assuming it will be time-dependent? I can foresee this would be practically difficult. How about just using beta-binomial for all groups in the future, with shared beta parameters across all groups? That way you enforce higher variability in all groups (which is consistent with overdispersion in the data) without overcomplicating the model. I’m not suggesting you make this change but discuss here as a possible future improvement.

Line 629: why stop at piecewise regression for selection coefficients, and not take on even more flexible forms like additive splines or Gaussian processes? I think you need to also acknowledge the necessary drawbacks of a more complicated model, and defend the parsimony of your current model as a strength. – Ah I see you immediately discussed this in the following paragraph – I think you could consolidate these paragraphs in one to highlight why you chose to implement your model this way, and its relative merits.

Appendix: for easier use by readers, I suggest uploading your codes in a GitHub repository, instead of being formatted inside a pdf

Also consider reporting computation time - I am very impressed EM in R is even computationally feasible to run for a dataset this size, I imagine many other readers would want to know how scalable this is, especially in comparison to other existing implementations in R through underlying C++ codes.

Reviewer #2: The manuscript proposes a novel unsupervised method that clusters the trajectories of viral mutation frequencies from wastewater samples and estimates associated parameters to facilitate early detection and fitness assessment of emerging SARS-CoV-2 variants. This approach could have significant implications for future responses to public health emergencies. Overall, I find the work innovative and promising. However, I have several major and minor comments:

Major Comments:

1. Assumption of Independence in the Joint Probability Expression (Page 8, Line 137): The manuscript appears to assume that observations across different time points are independent. However, real-world time-series data often exhibit clear temporal dependencies. I recommend the authors either provide theoretical justification or include additional simulation studies to examine how violations of this independence assumption could affect the estimation results. Such analysis will significantly enhance the method's practical applicability.

2. Parameter Identifiability: In the proposed likelihood function, there is potential correlation among parameters (e.g., μ and s or others). One challenge in the estimation procedure is ensuring the identifiability of these parameters. I recommend that the authors explicitly discuss parameter identifiability, either theoretically or through targeted simulation studies, to clarify under which conditions these parameters are identifiable.

3. Effectiveness of the Two-Step Initialization Strategy: The authors propose a two-step initialization strategy to mitigate the sensitivity of the EM algorithm to initial values, which is commendable. However, I would appreciate further discussion regarding its effectiveness with more complex data distributions or larger datasets. Providing insights from the authors' experience on when this initialization strategy is effective and when additional measures might be necessary (without requiring a new explicit strategy) would greatly clarify the generalizability and robustness of the proposed method.

Minor Comments:

4. Figure 1 Presentation: The current layout of Figure 1, with two subplots placed side by side, appears crowded, and the x-axis dates are overlapping. Additionally, the figure should clearly explain what the different colors represent. At present, the figure does not seem to convey sufficient useful information and may need reformatting for clarity.

5. Algorithm Pseudocode: To facilitate better understanding and quick comprehension of the proposed algorithm, I suggest including pseudocode or a flowchart in either the main text or the appendix.

6. Availability of Code: Currently, the code files are not available through the provided GitHub repository (although some snippets are present in the appendix). To further improve the transparency and reproducibility of this research, I strongly encourage the authors to publicly release the code and provide detailed README documentation, enabling other researchers to easily reproduce the experimental results.

Overall, the manuscript presents a method with significant innovation and practical potential. I look forward to the authors carefully addressing these points to enhance the paper's persuasiveness and broad applicability.

Reviewer #3: Lefebvre and colleagues set out to develop a method to detect clusters (groups) of mutations with similar temporal trajectories, in order to detect SARS-CoV-2 variants in wastewater genomic samples. The method is cleverly designed and rigorously tested. The method has a clear downside in that it lacks clear criteria for determining the number of groups of mutations in the data, as the statistics favour eve smaller groups of mutations. This can be a particular problem when trying to detect newly emerging variant in real time extracted wastewater surveillance data. The danger here, is that when employing such a method in real-time,

However, to the authors credits, they do discuss this important limitation, and thus set the scene for future improvements of this, and related, methods.

In my opinion, however, the great weakness of the paper is in the writing. The authors performed a quite comprehensive analysis of the method’s performance and wrote it down in great detail. As a result, the paper is very long, and very hard to follow at times. I had trouble getting through the paper, despite my interest in the topic. If the authors want to reach a larger audience, I’d suggest making numerous changes to the manuscript in order to increase its readability.

1) Remove any data from the text that is also mentioned in figures or tables.

On numerous occasions, the authors present information in the text that is also mentioned in tables. A clear example of this is the second paragraph of the results (line 171 “setting K =1 non-neutral group… ” until “in table 2.”, line 181). This information is also available in table 1, and even added to table 2, and therefore available to the reader, but makes the paragraph very hard to read. This happens again on page 13, from line 203 onwards, as well as line 331 (page 22), line 407 (page 28), line 422 (page 29), and line 530 (Page 38). There are probably more places where information can be removed from the text and just referred to the table to improved the flow of the text.

2) Don’t reference in forward direction.

The authors have a tendency to refer to sections of the paper that the reader hasn‘t read yet. In particular, to section 4 (Discussion and perspectives). This is rather disruptive to the reader, as it feels like from there on, vital information is missing when following the paper chronologically, because the Discussion hasn‘t been read.

Make sure that the presentation of results is done in such a way that it follows logically to the reader. The results section presents the results of the analyses proposed in the methods section, and the reader expects their implications to be discussed in the discussion section. There is certainly room for interpretation in the results section, but if the narrative is consistent, the reader can wait for wider implications to be discussed later on (in the Discussion section).

3) Refer to analyses and sections by name, not by letter.

Throughout the paper, different parts of the analysis are referred to by letter (e.g. “Simulation scheme A”, “Section 4”, “Analysis B and C”). This means that the reader has to go back into the text or tables to find out which parts are referred to. I would want to suggest to refer to all these parts with some name rather than the letters, to make it easier for the reader.

4) Move part of the analyses to a supplementary information

All in all, the text is rather long, and some of the sub-analyses seem repetitive (but still needed!). Similar simulation schemes could for instance be moved to the supplementary information, with their differences mentioned in the main text. Shortening the text will make it more accessible to a larger audience.

5) Be specific

In the methods section, the authors use “few iterations” twice, and “few time” once. Please be specific on how many times this was done. Currently this reads like “After fiddling around a bit, the initialisation magically worked.”. The same applies to the use of the “Elbow method”: it has no clear criteria for determining where the “elbow” actually is located, making it very subjective. If the method is to be used in real-time to detect variants, specific criteria for the selection of the number of clusters are needed.

I am left with a couple a small comments:

- Given that the method is first tested on simulated data, I would suggest moving the subsection on Data (2.1) to after the one on Mathematical Modeling (2.2).

- Page 6, line 125: the logit transformation of mutation frequencies is not purely practical for visual purposes. The expected trajectory of a fitness-conferring SNP is linear when logit-transformed.

- A matter of taste: I would suggest renaming the section “Real data” to Wastewater data”, to avoid any discussion on what “Real” means.

- figure 6 says “log frequency”, instead of “logit frequency” in the legend for panel (b)

- Please note that “public health strategies” (page 26, line 363) are not expected to influence selection, and with that the mutation’s frequency trajectory, because PH strategies generally influence all variants equally in the absence of methods to distinguish between cases caused by different variants in the field in real-time.

**Have the authors made all data and (if applicable) computational code underlying the findings in their manuscript fully available?**

Reviewer #1: Yes

Reviewer #2: **No: **The dataset is publicly available, but I did not find the code in the GitHub link provided by the authors (although some snippets are present in the appendix). The author needs to confirm the availability of the link again.

Reviewer #3: None

PLOS authors have the option to publish the peer review history of their article (what does this mean?). If published, this will include your full peer review and any attached files.

Reviewer #1: **Yes: **Tianxiao Hao

Reviewer #2: No

Reviewer #3: **Yes: **Tjibbe Donker

**Figure resubmission:**
---

## [Decision Letter · Decision Letter 1]

22 Oct 2025

PCOMPBIOL-D-25-00088R1

Unsupervised detection and fitness estimation of emerging SARS-CoV-2 variants: Application to wastewater samples (ANRS0160)

PLOS Computational Biology

Dear Dr. Lefebvre,

Thank you for submitting your manuscript to PLOS Computational Biology. After careful consideration, we feel that it has merit but does not fully meet PLOS Computational Biology's publication criteria as it currently stands. Therefore, we invite you to submit a revised version of the manuscript that addresses the points raised during the review process.

Please submit your revised manuscript within 60 days Dec 22 2025 11:59PM. If you will need more time than this to complete your revisions, please reply to this message or contact the journal office at ploscompbiol@plos.org. Please include the following items when submitting your revised manuscript:

We look forward to receiving your revised manuscript.

Kind regards,

Andrey Rzhetsky

Academic Editor

PLOS Computational Biology

Roger Kouyos

Section Editor

PLOS Computational Biology

**Reviewers' comments:**

Reviewer's Responses to Questions

Reviewer #1: Dear authors,

Thank you for the opportunity to re-review the manuscript titled ‘Unsupervised detection and fitness estimation of emerging SARS-CoV-2 variants: Application to wastewater samples (ANRS0160)’ and well done on your extensive effort in improving the manuscript. In my view my comments and concerns in previous review have been adequately addressed, and I think you have sufficiently responded to the other reviewers as well. I regard this version of the manuscript as much improved in its clarity, level of detail, reproducibility, and scientific merit. However, I believe there is still some work to be done in terms of the overall narrative, particularly in clarifying the methodological approach and in highlighting the most critical findings, to maximise impact and relevance to readership of PLOS Comp Bio. My principal feedback is below:

First, although the rationale of your simulation studies and method for group number selection have been much improved in the main text, you now have a very long Method section, which in itself contains the results from the simulation study. This makes the overall narrative of your paper hard to follow as readers can get lost in the details in the methods. At the same time, the Method section, despite its length, has not sufficiently contextualised the need and the challenge of selecting groups. I propose the following changes: at the end of the ‘Mathematical modelling’ section, use some brief text to explain why you set up the model the way it is, and prompt the reader that the main implementation challenge is to simultaneously estimate the number of groups and group selection coefficients. Then, I suggest heavy-handed edits to condense the “Clustering and parameter estimation” – keep it short and clear that these are the necessary and complex approaches needed to solve these implementation challenges. I think you will have to move most of the existing text in this section to an appendix, and leave the main text as mostly a summary. Likewise, the ‘Model assessment over simulated datasets’ section needs to be condensed, and you need to be more explicit about its motivation, i.e., the nature of the model set up makes it challenging to identify group number and selection, so you ran a simulation study as proof of the robustness of your approach. I feel the same way about ‘The hidden random walk model’ section – it is good that you have explored the consequences of not explicitly encoding temporal correlation structure, but by the time the reader finishes this verbose section, they might become distracted or disoriented about the main message of the overall paper. I strongly recommend condensing the Methods section as much as possible, and especially pay attention not to introduce too many results of the simulation/random walk studies. In my opinion at most you could summarise the results of those complementary studies in text, but you should leave the figures to supplementary materials. You have too many figures as it stands now anyway.

There is a degree of dissonance between the Methods and the Results sections, which are both verbose and too long. While the Methods go into great details justifying the robustness of the approach, the Results likewise is very detailed in biological interpretations of the specific datasets, in their unique epidemiological contexts. These results are important, but it is confusing how they echo with the narrative in the Methods. For example, the comparison with supervised results, which in my opinion is crucial to the overall narrative, is not well anticipated in the Methods section. If you had just read the Methods section you do not get the impression that these Results would be reported. I admire the wide array of scientific queries you must have had to address to produce this work, but I worry this breadth is diluting the narrative cohesion of your manuscript, and as a result, is weakening its quality. Although it is clear all parts of the results are laborious to obtain, I think the way forward is to select the key message to present (which in my mind is how the unsupervised classification results compare against supervised ones, and what that tells us about the usefulness of your novel approach), and consolidate everything else into brief summaries, and provide full details in supplementary material.

To summarise, I believe the core scientific merit of this work is demonstrating how an unsupervised classification approach can uniquely provide early genomic shift signals in epidemic surveillance, including whether new strains have begun circulating and whether they have transmission advantage. This new approach is important in prompting timely response. The genomic surveillance over COVID pandemic and the unique epochs dominated by a succession of variants provided a unique data opportunity for the author team to test such an approach. However, implementing an unsupervised classification is hard work, and requires thorough stress-testing of both the model design and the implementation algorithm. I think at the moment the manuscript goes into too much detail about these implementation hurdles and loses the big picture of its impact in an epidemic response context.

As a side, I think this manuscript has too many figures, and the figures are needlessly complex. For example, the trendlines on Fig 8 are too messy to make sense of, and colour legend is missing – what is important to note on this figure? I do not think the data needs to be visualised just for visualisation sake, if it is so hard to make out a latent pattern from the plot. Likewise figures 10,11,13 have too many panels and the graphical elements are too small to see, you need to adjust line width, legend size etc for all of them. Once again I encourage thinking about which of those visualisations actually help with the narrative and which can be shelved into supp material.

Reviewer #3: The revised manuscript has improved considerably, and reads well. The authors have thoroughly addressed my previous comments, for which I thank them. I have no further comments.

**Have the authors made all data and (if applicable) computational code underlying the findings in their manuscript fully available?**

Reviewer #1: Yes

Reviewer #3: Yes

PLOS authors have the option to publish the peer review history of their article (what does this mean?). If published, this will include your full peer review and any attached files.

Reviewer #1: No

Reviewer #3: **Yes: **Tjibbe Donker

**Figure resubmission:**
---

## [Editor Report · Decision Letter 2]

17 Nov 2025

Dear Ms Lefebvre,

We are pleased to inform you that your manuscript 'Unsupervised detection and fitness estimation of emerging SARS-CoV-2 variants: Application to wastewater samples (ANRS0160)' has been provisionally accepted for publication in PLOS Computational Biology.

Best regards,

Roger Dimitri Kouyos

Section Editor

PLOS Computational Biology

Roger Kouyos

Section Editor

PLOS Computational Biology

---

## [Editor Report · Acceptance letter]

PCOMPBIOL-D-25-00088R2

Unsupervised detection and fitness estimation of emerging SARS-CoV-2 variants: Application to wastewater samples (ANRS0160)

Dear Dr Lefebvre,

I am pleased to inform you that your manuscript has been formally accepted for publication in PLOS Computational Biology. Your manuscript is now with our production department and you will be notified of the publication date in due course.

With kind regards,

Anita Estes
